# Transcriptome and Metabolome Analyses Reveal New Insights into the Regulatory Mechanism of Head Milled Rice Rate

**DOI:** 10.3390/plants11212838

**Published:** 2022-10-25

**Authors:** Wu Yang, Xianya Jiang, Yuelan Xie, Luo Chen, Junliang Zhao, Bin Liu, Shaohong Zhang, Dilin Liu

**Affiliations:** 1Rice Research Institute, Guangdong Academy of Agricultural Sciences, Guangdong Key Laboratory of New Technology in Rice Breeding, Guangdong Rice Engineering Laboratory, Guangzhou 510640, China; 2Yangjiang Institute of Agricultural Sciences, Yangjiang 529500, China

**Keywords:** rice, head milled rice rate (HMRR), differentially expressed genes (DEGs), metabolites

## Abstract

The head milled rice rate (HMRR) is the most important trait of milling quality, which affects the final yield and quality of rice. However, few genes related to HMRR have been identified and the regulatory mechanism of HMRR remains elusive. In this study, we performed a comparative analysis integrating the transcriptome sequencing of developing seeds at the grain-filling stage and a metabolome analysis of brown rice between two groups of accessions with contrasting performances in HMRR. A total of 768 differentially expressed genes (DEGs) were identified between the transcriptome profiles of low-HMRR and high-HMRR accessions. In comparison to the high-HMRR accessions, 655 DEGs were up-regulated in the low-HMRR accessions, which was 4.79 folds higher than the number of down-regulated genes. These up-regulated DEGs were enriched in various metabolic and biosynthetic processes, oxidation reduction, phosphorylation, ion transport and ATP-related processes. However, the 113 down-regulated DEGs in the low-HMRR accessions were concentrated in carbohydrate metabolic processes, cell-death-related processes and defense response. Among the 30 differential metabolites, 20 and 10 metabolites were down-/up-regulated, respectively, in the accessions with low HMRR. In addition, 10 differential metabolites, including five metabolites of the shikimate pathway and five metabolites of the pyruvate pathway, were integrated into two separate pathways, starting from sucrose. Our global analysis of HMRR provides an invaluable resource for a better understanding of the molecular mechanism underlying the genetic regulation of HMRR.

## 1. Introduction

Rice is one of the most important food crops, feeding more than half of the world’s population. With the green revolution and widespread use of hybrid rice, rice yields have been significantly improved in the past half century. At present, with the improvement in people’s living standards, high-quality rice is a foremost consideration for rice breeders and consumers. Rice quality mainly includes milling, appearance and cooking quality. Milling quality is evaluated by the brown rice rate (BRR), milled rice rate (MRR) and head milled rice rate (HMRR) [1]. Brown rice is produced by de-hulling rice by removing the lemma and palea. Milled rice is the product of brown rice after removing the bran. Head-milled rice is defined by a kernel equal to or longer than 3/4 full length of a kernel. Among the three parameters of milling quality, HMRR is the most critical factor affecting the commercial value of rice [2]. Therefore, high HMRR is the key trait selected for released rice cultivars.

HMRR is controlled by multiple QTLs or genes, and it is affected by environmental, harvesting, drying and milling processes [2,3]. Understanding the genetic basis of HMRR can greatly improve the efficiency of molecular breeding. However, few genes related to rice milling quality have been isolated, especially on HMRR. *Chalk5*, a major locus regulating rice chalkiness rate, was cloned from natural variants and elevated expression of *Chalk5* resulted in significantly reduced HMRR [4]. In most studies using bi-parental populations to identify the QTL of HMRR, it is difficult to repeatedly detect the same QTL in different environments or years, indicating that HMRR is easily affected by the environment [2,5]. In recent years, genome-wide association studies (GWAS) with large germplasm resources have been used to identify the QTLs controlling milling quality in rice [6,7]. By using a diversity panel consisting of 272 *indica* accessions, six QTLs were detected for HMRR by GWAS and a stably expressed QTL (*qHMRR3.1*) was identified in the two tested environments [6]. Wang et al. [7] identified two QTLs of HMRR distributed on chromosomes 3 and 9 by GWAS in a panel of 258 accessions selected from the 3K Rice Genome Project, but these two QTLs could only be identified in one environment. Due to the complexity of HMRR [2,5,6,7], it is necessary to study the regulation mechanism of HMRR from a global perspective.

Many studies have shown that grain filling and endosperm development at the grain-filling stage are very important for rice-grain quality. Grain filling is a complex and orderly dynamic process. The assimilation products in leaves are transported to the storage organs (grains) in the form of sucrose, which is converted to starch or other metabolic reactions by a series of enzymes. The whole process is coordinated, balanced or antagonized by enzymes, hormones and genes, which jointly regulate the seed development and grain quality [8,9,10]. So far, there have been few global studies of grain filling in accessions with differential HMRR, which limits the overall understanding of the regulation mechanisms of HMRR at the grain-filling stage.

Transcriptome sequencing is a useful tool for obtaining gene-expression data and metabolites are the basis and direct reflection of an organism’s phenotype. In recent years, integrated transcriptome and metabolome analyses have been successfully applied to study various biological processes in plants [11,12]. In this study, the transcriptomes of developing seeds at the grain-filling stage and the metabolome of brown rice in two groups of accessions with contrasting HMRR were examined. In total, 768 differentially expressed genes (DEGs) and 30 differential metabolites between the two contrasting accessions were identified, respectively. The transcriptome results showed that more DEGs were up-regulated in the low-HMRR accessions, and that enhanced energy activities at 14 days after flowering and premature senescence at 21 days after flowering may be important causes of low HMRR. The integrated pathway of metabolites indicated that the shikimate and pyruvate metabolic pathways starting from sucrose may be related to HMRR. Our study provides new insights into the regulation mechanism of HMRR and facilitates the gene cloning of HMRR.

## 2. Results

### 2.1. Screening of Rice Accessions with Differential HMRR

Previous studies have shown that HMRR is easily influenced by the environment. In order to screen suitable rice accessions for the omics study, six *indica* accessions exhibiting stable phenotypes of HMRR in two environments were selected (Figure 1 and Table 1). The six accessions had almost the same heading date (Table 1). Among them, three accessions showed low HMRR (L accessions: L1~L3) in the range of 24.40~26.80%. The other three accessions showed high HMRR (H accessions: H1~H3) in the range of 64.90~67.10% (Table 1). In addition, due to the negative correlation between HMRR and grain shape [7], two accessions (H2 and H3) with relatively high grain length–width ratios but simultaneously showing high HMRR (Table 1) were selected for the follow-up study to prevent the interference of grain shape on HMRR.

### 2.2. Transcriptome Profiles of Developing Seeds at Grain-Filling Stage

During grain filling, the gene expression in developing seeds plays a key role in the formation of rice quality [8,10]. To compare the transcriptional landscape between the L and H accessions, we attempted to perform an RNA sequencing of the developing seeds at 14 and 21 days after flowering and evaluated the gene expression. Unexpectedly, the RNA of the developing seeds in the L accessions degraded significantly at 21 days after flowering, and the 28S RNA and 18S RNA were almost undetectable (Figure 2A). The RNA quality of the developing seeds at 21 days after flowering in the L accessions could not be used for subsequent library construction and sequencing. Therefore, only the RNA samples collected at 14 days after flowering were used for the transcriptome analysis.

In total, 768 DEGs in the developing seeds at 14 days after flowering were identified between the L and H accessions (Appendix A). Interestingly, the L accessions exhibited more actively expressed genes. Compared to the H accessions, 655 of the 768 DEGs were up-regulated in the L accessions and 113 DEGs were down-regulated, respectively (Figure 2B). To validate the results of the RNA sequencing, we randomly selected 10 genes for qRT-PCR. A correlation analysis showed that the qRT-PCR results were highly correlated with the transcriptome profiles, and the Pearson coefficient was 0.87 (Figure 2C). These results showed that the RNA sequencing data were reliable.

### 2.3. GO Enrichment Analyses of DEGs

GO analyses were performed for the DEGs. The annotations of the 655 DEGs highly expressed in the L accessions were concentrated in various metabolic and biosynthetic processes, oxidation reduction, phosphorylation, ion transport and ATP-related processes (Figure 3A), and the annotations of 113 DEGs actively expressed in the H accessions were concentrated in the carbohydrate metabolic process, cell-death-related processes and defense response (Figure 3B). These results suggest that at 14 days after flowering, energy synthesis and metabolic activity in the H accessions were depressed, while cellular programmed death was activated in the H accessions.

We further analyzed the major gene categories of the DEGs. Interestingly, the expression of 46 DEGs related to ATP metabolism and 5 DEGs related to ethylene-response factors in the H accessions were significantly lower than those in the L accessions (Figure 3C,D, Appendix A). Other gene categories, including starch synthesis and glucose metabolism (36 DEGs), amino acid metabolism and transport (49 DEGs), endosperm specific genes (22 DEGs), transcription factors (24 DEGs) and zinc finger protein (22 DEGs) did not show clear L- or H-accession-specific expression patterns (Appendix A).

At the grain-filling stage, the starch synthesis and accumulation in developing seeds is the major process contributing to grain quality. At the same time, starch hydrolysis also occurs; this process negatively affects grain quality. Among the 36 DEGs related to starch synthesis and glucose metabolism, 9 genes were highly expressed and 27 genes were repressed in the H accessions. Eight hydrolase genes (*LOC_Os01g04290*, *LOC_Os01g71474*, *LOC_Os04g40510*, *LOC_Os04g45290*, *LOC_Os04g51460*, *LOC_Os06g46284*, *LOC_Os11g18730*, *LOC_Os11g47520*) in the H accessions were expressed at a very low level, while the other four hydrolases were expressed at a high level (*LOC_Os01g71380*, *LOC_Os05g30280*, *LOC_Os10g28080*, *LOC_Os11g27400*) (Appendix A).

### 2.4. Metabolite Profiles of Brown Rice

Metabolites are not only the final products of gene transcription and translation, but they are also the basis and direct reflection of an organism’s phenotype. The activities of the DEGs between the L and H accessions may have resulted in different profiles of metabolite accumulation, which cumulatively resulted in the difference in HMRR. To investigate the differences in metabolite accumulation, a widely targeted metabolome method was used to quantify the metabolites in the L and H accessions of the brown rice. As a result, a total of 308 metabolites were detected by LC–MS (Appendix A). The PCA analysis showed that the L and H accessions were clearly separated in the PC1 × PC2 score plots (Figure 4A,B). A total of 30 metabolites showed significant differences between the L and H accessions, in which the accumulation of 20 metabolites was higher and the other 10 metabolites were lower in H accessions (Figure 4C, Appendix A). A further analysis showed that the 30 metabolites were mainly clustered in the metabolism processes of amino acids, lipids, cofactors and vitamins, carbohydrate, nucleotide and energy (Figure 4C).

### 2.5. Integrated Pathway Analysis of Differential Metabolites

During the grain-filling stage, sucrose is transported from leaves to developing seeds and becomes the most important substrate in starch synthesis and various metabolic reactions. In order to analyze the relationship between the differential metabolites and HMRR, we conducted an integrated analysis of these metabolites in the sucrose pathways. The results showed that 10 differential metabolites were integrated into two metabolic pathways of sucrose. The five down-regulated metabolites in the H accessions (3, 4-Dihydroxyphenylglycol, 3, 4-Dihydroxymandelic acid, 3-(3, 4-Dihydroxy-5-methoxy)-2-propenoic acid, Taxifolin, Epicatechin) were all in the shikimate metabolic pathway, while the five up-regulated metabolites in the H accessions (1-Methylhistidine, L-Arginine, Prostaglandin C2, Prostaglandin H2, 9-oxoODE) were all in the pyruvate metabolic pathway.

## 3. Discussion

During grain filling, metabolites such as sugars and amino acids are transported to seeds and involved in different biosynthetic pathways. The pathways must be regulated and coordinated so that the genes involved are expressed at proper levels at each stage of development [13]. In this study, we focused on the gene regulatory network and metabolic pathways controlling the HMRR of rice. HMRR is easily affected by environments [3,7]. Therefore, we used two groups of accessions showing stable but contrasting HMRR in different ecological environments for comparative omics studies. All six accessions had the same heading date to eliminate possible environmental interference. The grain shape was mainly determined at the booting stage, while HMRR was greatly affected by the grain-filling stage [8]. There was a negative correlation between the grain type and HMRR, which was caused by the fact that grains with higher grain length–width ratios are easier to break during processing [2,7]. The H accessions in this study had higher grain length–width ratios but simultaneously showed high HMRR (Table 1), which is more conducive to our study on the influence of gene expression in HMRR during grain filling. Therefore, comparative omics between the two contrasting groups can be more informative for the regulation mechanism of HMRR.

An earlier transcriptional comparison was performed between a high-milling-yield cultivar, Cypress, and a low-milling-yield cultivar LaGrue. A higher expression of the genes involved in starch metabolism in Cypress was observed during early (6-day-old) seed development [14]. In our study, the RNA-sequencing analysis showed that most of the DEGs (655 out of 768) of the developing seeds at 14 days after flowering were highly expressed in the L accessions, indicating that the gene expressions in the L accessions were more active. The 655 DEGs up-regulated in the L accessions were concentrated in various metabolic and biosynthetic processes, oxidation reduction, phosphorylation, ion transport and ATP-related processes. It is noteworthy that the expression levels of 46 DEGs related to ATP metabolism were significantly higher in the L than those in the H accessions (Figure 3C). Furthermore, more hydrolase genes were actively expressed in the L accessions (Appendix A), which corresponded with the stronger metabolic activities of the ATP in the L accessions, suggesting that the developing seeds of the L accessions at this stage might have been deficient in energy or needed to synthesize more ATP for the present life process.

Seed development is closely related to senescence [15]. The RNA of the developing seeds in the H accessions maintained high quality at 21 days after flowering, while the RNA of the developing seeds in the L accessions was severely degraded (Figure 2A), indicating that premature senescence is more intense in L accessions. The L accessions used in this study showed very extreme phenotypes, with HMRR as low as 24.40~26.80%. These extreme phenotypes indicated that reduced HMRR may be due to enhanced RNA degradation and premature senescence. So far, there have been many studies on leaf senescence in rice, and it is agreed that breeding anti-senescence rice varieties is of great significance for the stability and improvement of rice yields [11,15]. Grain filling is a complex and orderly dynamic process. During aging, the assimilation products in leaves are transported to grains in the form of sucrose, which is converted to starch or other metabolic reactions by a series of enzymes. The coordination and source (leaf)–sink (grain) balance are very important for seed development in rice [8,9,10]. The premature senescence of developing seeds in L accessions may break the source–sink balance, which results in decreased HMRR. In addition, ethylene is an important plant hormone related to senescence. Previous studies have shown that the ethylene-evolution rate in grains during the grain-filling period correlates negatively with HMRR [16,17]. In the present study, five DEGs of ethylene-response factors in L accessions were significantly higher than those in H accessions (Figure 3D, Appendix A), suggesting that ethylene activities may be one reason for the differential HMRR. Highly expressed ethylene-response factors and activated ethylene pathway might cause low HMRR by accelerating the senescence in grains and the tissues involved in assimilate transport [17]. Altogether, rice-seed growth and development were controlled by multiple genes [18]. In particular, balanced energy activities at 14 days after flowering and tightly controlled premature senescence at 21 days after flowering may be critical for the maintenance of high HMRR in rice.

To further investigate the regulation mechanism of HMRR on the metabolite level, we identified 30 differential metabolites between the two contrasting groups of accessions (Figure 4C). Interestingly, 10 differential metabolites were integrated into two separate pathways of sucrose (Figure 5), suggesting metabolically biased pathways of accessions with high and low HMRR. Therefore, the shikimate and pyruvate metabolic pathways starting from sucrose may be related to HMRR. So far, few studies have focused on the relationship between metabolites and rice quality, especially HMRR. Our study provides new insights into the regulatory mechanism of HMRR from the perspective of metabolites and metabolic pathways.

## 4. Materials and Methods

### 4.1. Plant Materials and Growth Conditions

The six rice accessions used in this study were selected from a subset of the RDP2 consisting of 442 rice accessions, which were planted in the experimental fields of Guangzhou and Yangjiang in Guangdong Province, China, in the second cropping season in 2016 and 2018, respectively. The field management, including irrigation, fertilization and pest control followed the traditional rice production practice. Guangzhou features a subtropical monsoon climate, while Yangjiang features a subtropical maritime monsoon climate. The influence of light and temperature differences between the two environments on growth period and pericarp color of rice was mentioned in our previous study [19].

### 4.2. Evaluation of HMRR

The HMRR was measured according to the National Standards of the People’s Republic of China (GB/T 21719-2008). Briefly, The seeds were dehulled by an electrical dehuller (JLG-II, Chengdu, China) and milled by a rice miller (JNM-II, Chengdu, China). The grain weight (M_0_) and head-milled-rice weight (M_1_) were measured for each rice accession. The HMRR was calculated as ((M_1_/M_0_) × 100%).

### 4.3. RNA Extraction and Quality Detection

On the 14th and 21th day after flowering, three panicles with the same flowering time and consistent growth were sampled from each accession as three independent replicates. Subsequently, RNA extraction and detection were carried out under the same experimental conditions. The total RNA was extracted using Trizol reagent (Takara, Dalian, China). Each accession consisted of three biological replicates. The concentration and integrity of RNA were detected using Agilent 2100 Bioanalyzer (Agilent, Palo Alto, CA, USA) and Agilent RNA 6000 Nano Kit (Agilent, Palo Alto, CA, United States).

### 4.4. RNA Sequencing and GO Analyses

RNA sequencing was performed by the Annoroad Gene Technology (Beijing, China), and data analysis was conducted as described in our previous study [19]. The DEGs between the two sets of contrasting accessions were identified according to the criteria of *p*-value ≤ 0.01 and fold change of pairwise comparison ≥1.5 or ≤0.67. Gene-ontology (GO) analysis was performed through AGRIGO2 (http://systemsbiology.cau.edu.cn/agriGOv2/) on 1 to 10 June 2020.

### 4.5. Real-Time PCR Analysis

The RNA samples for RNA-sequencing assays were used to confirm the results of RNA sequencing. Real-time PCR was conducted using the same method as described by Yang et al. [20]. The qRT-PCR analysis was performed using the CFX 96 system (Biorad, Hercules, CA, USA). The primers were designed by the Primer designing tool on NCBI. The gene-specific primers are listed in Appendix A. The *EF1α* was used as the normalization gene of mRNA. All reactions were repeated three times.

### 4.6. Metabolite Identification and Data Analysis

The extraction of metabolites was based on published methods [21,22]. Chromatographic separation was accomplished in a Thermo Ultimate 3000 system (Thermo, Waltham, MA, USA) column. The ESI-MSn experiments were executed on the Thermo Q Exactive Focus mass spectrometer with the spray voltage of 3.8 kV and −2.5 kV in positive and negative modes, respectively. The metabolic profiling procedures were conducted according to [23]. The identification of the metabolites was according to the metabolite database established by BioNovoGene Corp. (Suzhou, Jiangsu, China). The differential metabolites between the two sets of contrasting accessions were identified according to the criteria of *p*-value ≤ 0.05 and fold change of pairwise comparison ≥1.5 or ≤0.67. Pathway analysis was conducted using Metaboanalyst 3.0, mainly based on KEGG pathway.

### 4.7. Integration Analysis of Metabolite Pathways

According to metabolite pathways of KEGG database, combined with pathways related to glucose metabolism (glycolysis and TCA cycle), amino-acid metabolism, arachidonic-acid metabolism and flavonoid metabolism, the differential metabolites were displayed in related pathways. The integrated pathway of metabolites was mapped manually.

## Figures and Tables

**Figure 1 plants-11-02838-f001:**
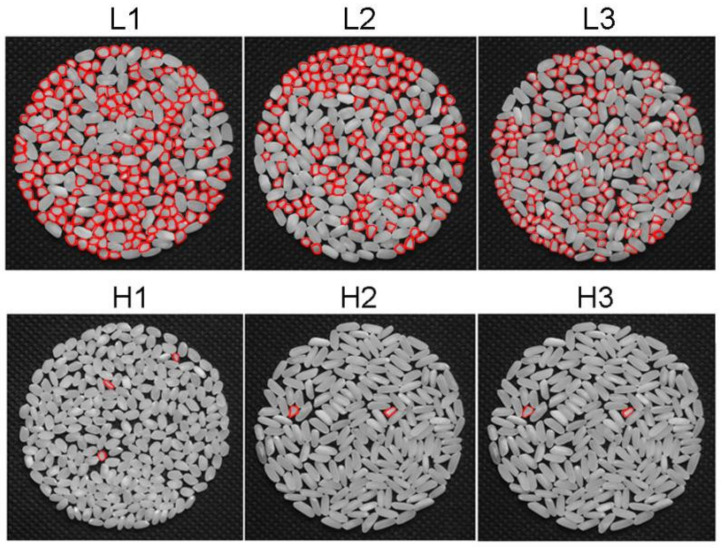
The HMRR phenotypes of L and H accessions. In total, 20 g of grains of each accession were dehulled by an electrical dehuller and then milled by a rice miller. The same weight of milled rice was used for the photograph. The broken milled rice are marked with red lines.

**Figure 2 plants-11-02838-f002:**
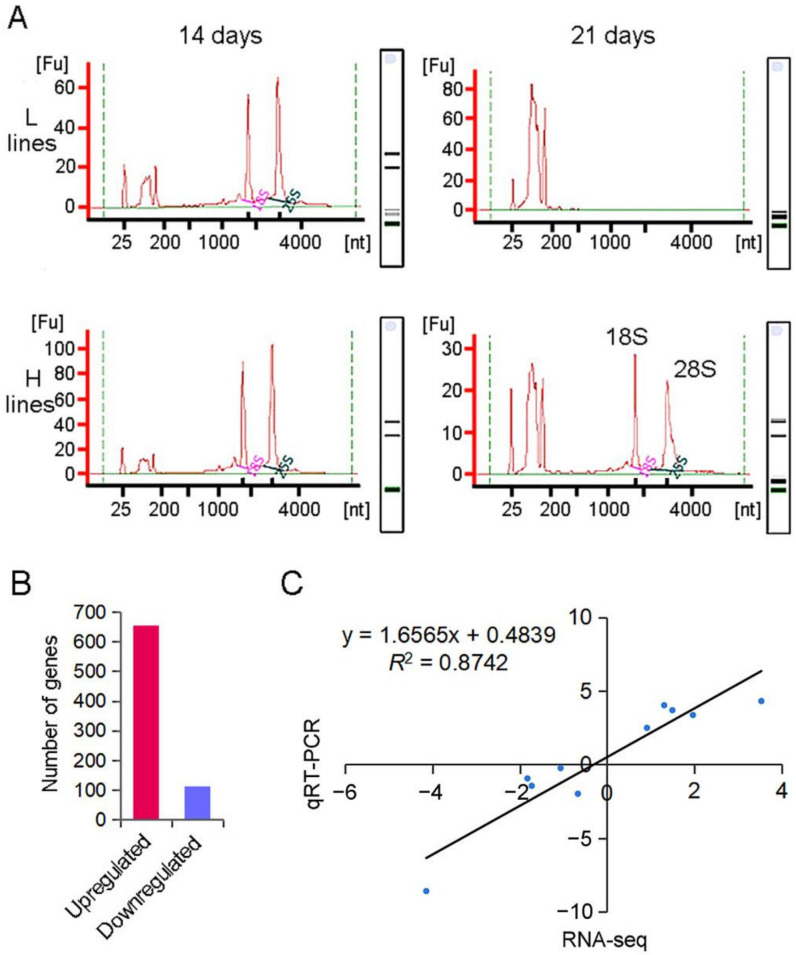
Comparison of RNA quality and DEGs between L and H accessions. (**A**) Comparison of RNA quality of developing seeds between L and H accessions at 14 and 21 days after flowering. nt: nucleotide. Fu: fluorescence value. (**B**) The number of genes up-regulated and down-regulated in L accessions. (**C**) Regression of qRT-PCR on RNA-sequencing DEGs. The correlation coefficient (*R*^2^) is indicated in the figure. DEGs: differentially expressed genes.

**Figure 3 plants-11-02838-f003:**
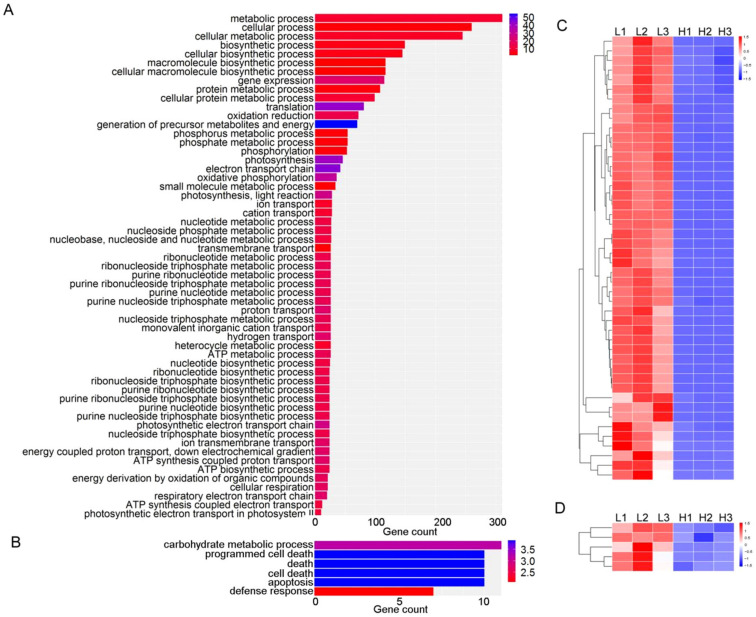
GO enrichment analyses of DEGs. (**A**,**B**) GO enrichment analyses of DEGs up-regulated in L and H accessions, respectively. (**C**,**D**) The heat maps of 46 DEGs related to ATP metabolism and 5 DEGs of ethylene-response factors, respectively.

**Figure 4 plants-11-02838-f004:**
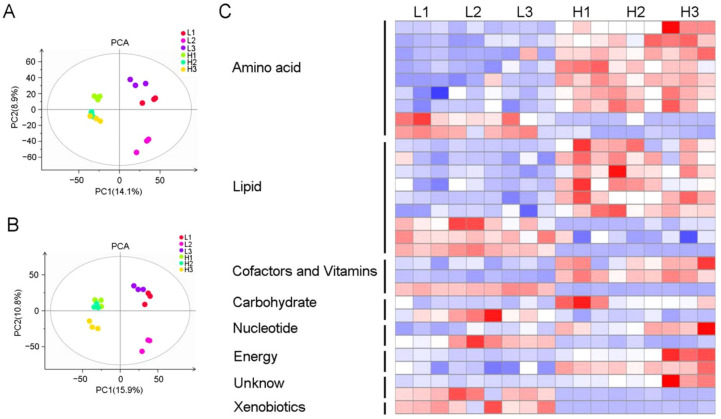
Metabolic analysis of brown rice in L and H accessions. (**A**,**B**) The PCA of all samples in positive- and negative-ion mode, respectively. PCA: principal component analysis. (**C**) KEGG classification and heat map of 30 differential metabolites.

**Figure 5 plants-11-02838-f005:**
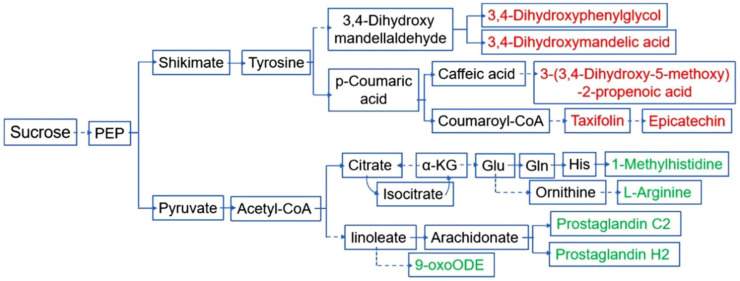
Integrated pathway of differential metabolites in L accessions. The red and green characters represent up- and down-regulated metabolites, respectively, in L accessions compared to H accessions. Solid lines represent direct reactions, dotted lines represent indirect reactions.

**Table 1 plants-11-02838-t001:** Information on the six rice accessions used in this study.

No.	Country of Origin	HMRR (%)	Heading Date (d)	Grain Length (mm)	Grain Width (mm)	Grain Length–Width Ratio
L1	Bangladesh	24.70 ± 4.67	67.83 ± 3.06	7.99 ± 0.16	3.85 ± 0.05	2.07 ± 0.01
L2	Bangladesh	24.40 ± 5.75	64.42 ± 1.53	7.84 ± 0.15	3.46 ± 0.05	2.27 ± 0.01
L3	India	26.80 ± 6.64	72.92 ± 3.65	8.49 ± 0.16	3.90 ± 0.06	2.18 ± 0.01
H1	Bangladesh	67.10 ± 1.20	70.33 ± 3.77	6.06 ± 0.11	3.46 ± 0.05	1.75 ± 0.01
H2	Pakistan	66.98 ± 1.55	71.67 ± 1.89	9.04 ± 0.17	3.40 ± 0.05	2.66 ± 0.01
H3	Pakistan	64.90 ± 2.83	70.42 ± 1.53	9.70 ± 0.18	2.60 ± 0.04	3.73 ± 0.02

## Data Availability

All data generated or analyzed during this study are included in this published article and its Appendix A.

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
