# Peer review of "Transcriptome and Metabolome Analyses Reveal New Insights into the Regulatory Mechanism of Head Milled Rice Rate"

_plants, 2022, doi:10.3390/plants11212838_

Round 1
Reviewer 1 Report
Dear Authors,
The manuscript presented looks good. I have few questions and suggestions for the improvement of the manuscript and would like you to include it in the text.
1- Very few references have been cited in all section of the paper. Most of the references are older than 2016. Only 5 references from 2020-21.
2-Introduction section- please elaborate on rice as an important crop.
3- GWAS- no reference. Please elaborate and add references.
4- Line 51-Add references.
5- Discussion section looks weak and does not support the claim of author. Mostly repetitive of results. Few references added in discussion section.
6- Figure 1A- Instead of image 1A please present a table discussing length and with of all selected varieties.
Author Response
The manuscript presented looks good. I have few questions and suggestions for the improvement of the manuscript and would like you to include it in the text. 1-Very few references have been cited in all section of the paper. Most of the references are older than 2016. Only 5 references from 2020-21. Answer: Thanks a lot for the reviewer’s valuable comments. We fully agree that adding some new references will improve the quality of the manuscript. Therefore we searched again for references on rice milling quality. Though genetic studies on milling quality, especially for HMRR are rarely reported, we were able to find some related references. Eventually we added 2 more references, which are relatively new and closely related to our work. 2-Introduction section- please elaborate on rice as an important crop. Answer: We have added some new descriptions in the “Introduction section” (line 32-36). 3-GWAS- no reference. Please elaborate and add references. Answer: We have elaborated and added the references (line 52-59) accordingly as suggested by the reviewer. 4- Line 51-Add references. Answer: We have added references (line 59). 5- Discussion section looks weak and does not support the claim of author. Mostly repetitive of results. Few references added in discussion section. Answer: Thanks a lot for the comments. Omics studies on HMRR have not been reported, and there are few genes and reports related to HMRR. Due to the complexity of HMRR, it is necessary to study the regulation mechanism of HMRR from a global view. In this study, we focused on the influence of gene expression on HMRR during grain filling. To better understand the observation we’ve made in the results, we added some detailed discussions in the revised manuscript, and we hope that the existing results of this study can bring enlightenment to relevant researchers (line 174-180, 204-215). Accordingly, references were cited in discussion section to support our claim. 6- Figure 1A- Instead of image 1A please present a table discussing length and width of all selected varieties. Answer: We have adjusted Figure 1 and added a table (Table 1) to demonstrate the agronomic features of the six materials in more detail. Meanwhile we describe the sources of the six materials and the differences between the two environments tested in “4.1. Plant Materials and Growth Conditions”. We also added discussion on the relationship between grain shape and HMRR of the six accessions used in this study (line 174-180).Reviewer 2 Report
Transcriptome and metabolome analyses reveal new insights into the regulatory mechanism of head milled rice rate
Yang et al, submitted to Plants 2022
The manuscript describes the results of a set of analyses between two contrasted subsets of rice accessions for head milled rice rate, including a transcriptomics approach through RNA sequencing and an untargeted metabolomics approach. Thus, the authors identified sets of differentially expressed genes as well as differentially expressed metabolites. Functional enrichment and consistency between the two approaches led the authors to conclude on the importance of energetic pathways starting from sucrose in the differentiation of low and high head milled rice rate.
The manuscript is relatively well-written with a clear and easy to follow organization. The article could be of broad interest, both for researchers interested in the genetics and biology behind the final phenotype of rice grain quality, but also to researchers wishing to integrate different omics data. However, it suffers from several issues that need to be addressed:
- It was not clear how the plant materials were used as the authors say in line 77/78 “six indica accessions exhibiting stable phenotype of HMRR in two environments were selected” but only show the results after the selection. What sort of collection was screened to find and extract these 6 lines, what type of materials are they (recent elite material, older germplasm collected in genetic resources centre… the actual name is not necessary, but a minimum of information would be good)? Also, what were the differences between the 2 environments tested, were they representative of contrasted conditions?
- The authors further precise in lines 82/85, having selected lines with high HMRR, but contrasted for grain shape. This obviously was meant to ensure that all types of grain shapes were considered, but then it is not mentioned anywhere else in the manuscript. Should we understand that no link was found between grain shape and the biological pathways involved for HMRR?
- In the transcriptomics approach, the authors make the surprising observation that samples taken after 21 days in the LMRR accessions had very degraded RNA. The materials and methods section, as well as the results section are quite brief on this. We do not know how much materials were collected per biological sample and whether this observation was obtained repeatedly or if somehow, the RNA extraction for this particular batch failed. As only the samples after 14days were considered in the rest of the manuscript, I would recommend discarding all mention of the 21 days samples unless a clear explanation could be brought forward.
- To strengthen the point of the authors, could it be possible to link up the DEG with the DEM? Were there any DEGs that are known to be involved in the two main metabolic pathways pointed at by the authors? It seems the article is missing out on a concluding paragraph that would bring light on this.
Minor corrections include:
Page 3, line 96: “was” should read “were”
Pages 4, Figure 1A: the legend is not very explicit, what do the authors mean by “red box” ?
Author Response
The manuscript is relatively well-written with a clear and easy to follow organization. The article could be of broad interest, both for researchers interested in the genetics and biology behind the final phenotype of rice grain quality, but also to researchers wishing to integrate different omics data. However, it suffers from several issues that need to be addressed: -It was not clear how the plant materials were used as the authors say in line 77/78 “six indica accessions exhibiting stable phenotype of HMRR in two environments were selected” but only show the results after the selection. What sort of collection was screened to find and extract these 6 lines, what type of materials are they (recent elite material, older germplasm collected in genetic resources centre… the actual name is not necessary, but a minimum of information would be good)? Also, what were the differences between the 2 environments tested, were they representative of contrasted conditions? Answer: We have adjusted Figure 1 and added a table (Table 1) to show the information of the six materials in more detail. Meanwhile we describe the sources of the six materials and the differences between the two environments tested (line 255-263). -The authors further precise in lines 82/85, having selected lines with high HMRR, but contrasted for grain shape. This obviously was meant to ensure that all types of grain shapes were considered, but then it is not mentioned anywhere else in the manuscript. Should we understand that no link was found between grain shape and the biological pathways involved for HMRR? Answer: The grain shape is mainly determined at booting stage, and HMRR is greatly affected by grain filling stage. There is a negative correlation between the grain type and HMRR, which is caused by the fact that grains with higher grain length-width ratio are easier to break during processing. The H accessions in this study have higher grain length-width ratio but simultaneously showing high HMRR, which is more conducive to our study on the influence of gene expression on HMRR during grain filling. We have added relevant descriptions in the discussion section (line 174-180). -In the transcriptomics approach, the authors make the surprising observation that samples taken after 21 days in the L HMRR accessions had very degraded RNA. The materials and methods section, as well as the results section are quite brief on this. We do not know how much materials were collected per biological sample and whether this observation was obtained repeatedly or if somehow, the RNA extraction for this particular batch failed. As only the samples after 14days were considered in the rest of the manuscript, I would recommend discarding all mention of the 21 days samples unless a clear explanation could be brought forward. Answer: On the 14th and 21th day after flowering, three spikelets with the same flowering time and consistent growth were sampled from each accession as three independent replicates. Subsequent RNA extraction and detection were carried out under the same experimental conditions. Thus the RNA degradation in L accessions at 21th after flowering is not due to experimental manipulation. We have added the method descriptions (line 271-274). Because the L accessions used in this study were very extreme, which showed low HMRR in the range of 24.4%~26.8%. These extreme phenotypes indicates that reduced HMRR may be due to enhanced RNA degradation and premature senescence. We have added some content in the discussion section (Line 204-215). -To strengthen the point of the authors, could it be possible to link up the DEG with the DEM? Were there any DEGs that are known to be involved in the two main metabolic pathways pointed at by the authors? It seems the article is missing out on a concluding paragraph that would bring light on this. Answer: Thank you for the good question and kind suggestion. We used gene expression and metabolites pathways to try to link up the DEG with the DEM. Unfortunately, no DEGs identified in this study were involved in the two main metabolic pathways. This may be due to the fact that the DEGs were identified during grain filling, while the DEMs were identified after seed maturation. Subsequent studies on metabolites pathways starting from sucrose and contemporaneous gene expression should be a worthy consideration for HMRR. Minor corrections include: Page 3, line 96: “was” should read “were” Answer: Sorry for our carelessness. We have corrected it in the revision. Pages 4, Figure 1A: the legend is not very explicit, what do the authors mean by “red box” ? Answer: We have redescribed the legend of Figure 1.Round 2
Reviewer 1 Report
Dear Authors,
Thank you for including the suggested changes. The quality of the manuscript has improved.